# LEARNING EPIPOLAR FEATURE FIELDS FOR MULTI-IMAGE SUPER-RESOLUTION

## ABSTRACT

Multi-image super-resolution (MISR) allows to increase the spatial resolution of a low-resolution (LR) acquisition by combining multiple images carrying complementary information in the form of sub-pixel offsets in the scene sampling, and can be significantly more effective than its single-image counterpart. Its main difficulty lies in accurately registering and fusing the multi-image information. Currently studied settings, such as burst photography, typically involve assumptions of small geometric disparity between the LR images and rely on optical flow for image registration. We study a MISR method that can increase the resolution of sets of images acquired with arbitrary, and potentially wildly different, camera positions and orientations, generalizing the currently studied MISR settings. Our proposed model, called EpiMISR, moves away from optical flow and explicitly uses the epipolar geometry of the acquisition process, together with transformer-based processing of radiance feature fields to substantially improve over state-of-the-art MISR methods in presence of large disparities in the LR images.

## 1 INTRODUCTION

Image super-resolution (SR) is the task of recovering a high-resolution (HR) version of an image from degraded low-resolution (LR) observations. It is a longstanding inverse problem in the imaging field and has numerous practical applications due to camera limitations and image acquisition conditions. Most of the literature focuses on estimating the HR image from a single input image (SISR). While recent deep learning approaches (Liang et al. (2021), Kong et al. (2022), Chen et al. (2023)) have tremendously advanced the state of the art, SISR remains highly ill-posed due to the limited high-frequency information available in a single image. Multi-image SR methods (MISR), on the other hand, are presented with multiple samplings of a given scene, carrying complementary information at a sub-pixel level. MISR techniques seek to accurately fuse the multiple LR images to obtain SR images with significantly higher quality than what is achievable by SISR methods. Only recently the deep learning literature has started exploring the multi-image setting due to increased difficulty in creating benchmark datasets as well as developing effective methods that can handle accurate image registration.

MISR can be seen as a generalization of the classic Stereo-SR setting (Chu et al. (2022)), in which a pair of images is captured, often with a tightly controlled geometry to simplify the fusion process. At the moment, the most studied MISR settings are in the context of video (Wang et al. (2020)) where successive frames provide the multiple images, remote sensing images (Molini et al. (2019)) where satellite revisits of the same scene are exploited, and burst photography, where a set of photos is acquired in rapid succession such in Bhat et al. (2021), Lecouat et al. (2021) or Luo et al. (2022). All these settings present a common denominator in that variations in the acquisition geometry among the multiple images are relatively small, resulting in relatively small disparities in the image pixels. For example, in burst SR, geometric variations are mostly due to natural hand shaking. This is desirable because the SR process requires subpixel shifts in the sampling grid, and obtaining them with minimal overall movement only simplifies the fusion process. For this reason, works in this field resort on using forms of optical flow estimation between LR images to accurately register them. Optical flow estimates a translation vector for each pixel of an image in order to warp it to a target image. Such a transformation between flat camera planes may struggle in presence of complex 3D transformations.

It is thus clear that the aforementioned small-parallax settings that have been currently studied are restrictive and do not allow to account for many interesting scenarios for super-resolution where the LR images come from cameras with wildly different positions and orientations. As examples, one can think of sets of security cameras which image a scene from significantly different vantage points, or sets of images of a scene collected in the wild with no control over the acquisition process.

In this paper, we study the general MISR setting where a set of LR images are acquired by cameras with arbitrary positions and orientations, and our task is to super-resolve one (or more) of them. We move away from the optical flow based models, in favour of an explicit use of epipolar geometry with techniques inspired by recent works in the NeRF literature (Mildenhall et al. (2021)). However, contrary to the NeRF literature, we are not concerned with novel view synthesis, but rather follow the standard SR approach of restoring one of the observed LR images. Our proposed method, called EpiMISR, leverages strong spatial priors necessary for the SR task and transfomer-based processing of radiance feature fields to achieve effective fusion of images with large discrepancies in acquisition geometries. We show that EpiMISR substantially improves over the state-of-the-art SR techniques developed for the more restrictive scenarios.

## 2 RELATED WORK

### 2.1 SINGLE-IMAGE SUPER-RESOLUTION

Single image super-resolution (SISR) is a long-standing problem in the field of computer vision, aiming at recovering a high-resolution (HR) image $I^{\text{HR}}$ given its degraded version $I^{\text{LR}}$. In its simplest form, the forward model of the problem is:

$$I^{\text{LR}} = (K * I^{\text{HR}}) \downarrow_s \tag{1}$$

where $\downarrow_s$ denotes decimation by a factor $s$ and $*$ denotes a convolution with degradation kernel $K$.

Note that this problem is ill-posed as the degradation process is non-injective. To overcome this challenge, two main families of approaches have been proposed: regularization methods and data-driven methods. Regularizers such as total variation impose handcrafted a-priori knowledge to establish a criterion in order to choose a plausible SR image, as done by Babacan et al. (2008). Data-driven approaches, instead, extract this knowledge directly from data. Modern deep learning approaches to SISR descend from the pioneer works of Dong et al. (2015) and Lim et al. (2017). A recent state-of-the-art neural network design is SwinIR (Liang et al. (2021)) which leverages a windows-attention-based architecture. It is also worth mentioning that some works (Huang et al. (2020)) tackle the blind SISR problem, i.e., when the degradation process is not known and hence should be estimated. Finally, a branch of the literature is concerned with lightweight architectures, such as the one by Kong et al. (2022).

### 2.2 MULTI-IMAGE SUPER-RESOLUTION

The ill-posedness of SISR is intuitively reduced if extra images of the same scene are available. This MISR approach can be further specialized in the multiframe-SR if these extra images comes from adjacent frames of a video, burst-SR if they comes from a photo-burst, stereo-SR if the single extra image is the stereo companion of the target one.

Multiframe-SR and burst-SR assume small geometric disparity as there are small camera movements between successive acquisitions. Exploiting this fact, the first step in algorithms for these settings is typically to register the images to each other using optical flow models (Baker & Kanade (1999)). Recent works in the context of the burst-SR challenge by Bhat et al. (2022), such as Bhat et al. (2021), Lecouat et al. (2021), and Luo et al. (2022) follow this approach, relying on neural networks modules estimating optical flow. However, optical flow models geometric relations as locally translational on the camera plane, and, as such, is limited in its expressive power. This is fine when the geometric disparity is small, but a general setting may benefit for a more accurate account of the 3D geometry. On a similar note, lightfield SR Zhang et al. (2021) presents a grid-like and well-known arrangement of multiple cameras with small disparities and, therefore, allows simpler techniques for image fusion and does not have requirements of robustness as stringent as a large-disparity setting.

The stereo-SR setting, instead, assumes only the presence of two cameras (i.e., just one extra image) and the acquisition setting is typically controlled so that camera poses only differ by an horizon-

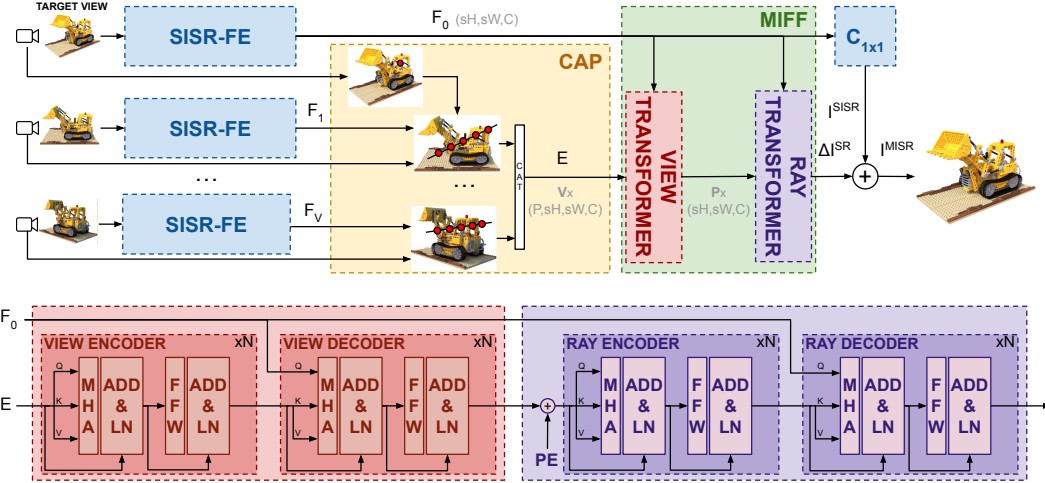

Figure 1: EpiMISR Architecture. From the LR target view and the extra views super-resolved features are obtained by any single-image SR network (SISR-FE), sampled along epipolar lines associated to pixels in the target view (CAP) and fused (MIFF) to produce a residual correction to single-image SR.

tal shift. Recently, Chu et al. (2022) developed a method for stereo-SR that utilizes an attention mechanism to perform image alignment implicitly.

To the best of our knowledge, this is the first work tackling the problem of generic multi-image super-resolution, i.e., there are no assumptions about the number of images or the relative poses of the cameras. Hence, we move away from 2D image alignment processes and leverage a full deep 3D world model.

## 2.3 NeRF AND IMAGE FUSION

NeRF architectures are neural world models, as they encode information from posed images in the weights of a neural network in a 3D-geometrically consistent way. In their original formulation by Mildenhall et al. (2021), a multilayer perceptron encodes the 5D radiance field of a given scene. Further evolutions, such as the works by Yu et al. (2021), Wang et al. (2021) and Huang et al. (2023) aim to avoid per-scene training, learning general priors by introducing a feature extractor and exploiting constraints from epipolar geometry in an explicit way. Barron et al. (2021), Isaac-Medina et al. (2023) and Barron et al. (2023) improve the ray casting procedure with cone casting and more advanced space sampling mechanisms. Varma et al. (2022), Suhail et al. (2022b) and Suhail et al. (2022a) move away from the physically-grounded volumetric rendering integral by replacing it with transformers acting on a feature space. Recently, NeRF-like models have also been used to address inverse problems in imaging. Pearl et al. (2022) and Mildenhall et al. (2022) address the case where the input views are noisy, discovering outstanding denoising performance. Wang et al. (2022a), Han et al. (2023), instead, tackle the problem of superresolving the NeRF 3D geometry model, hence being capable of generating novel-views at a higher resolution. Our work differs significantly from them in that we are concerned with super-resolution of existing views only and we do not optimize on a per-scene basis, but rather leverage a training set to train an image fusion model that can be then used for an arbitrary scene with an arbitrary number of views with an arbitrary geometry.

Some works, such as those by Huang et al. (2022), He et al. (2020) and Wang et al. (2022b), concerned with multi-image fusion leverage transformers in their pipelines. However, they differ from our work in that they do not deal with a super-resolution problem and are often limited by processing images in pairs and then aggregating the results with non-parametric processes.

## 3 PROPOSED METHOD

We address the setting in which a number of images of a given scene are acquired from arbitrary vantage points, possibly with large geometric disparity. These images have low resolution and we seek to super-resolve one of them by suitably combining the complementary information carried by the other images. Our proposed method, called **EpiMISR**, is a MISR neural network which explicitly accounts for the epipolar geometry by exploiting camera poses and processing 3D feature fields in a NeRF-like manner. Given $V + 1$ LR views of a static scene, and the corresponding intrinsic and extrinsic camera parameters, our task is to obtain a HR version of one of them, which we will call the *target view*, by also leveraging information from the $V$ extra views. In the parlance of NeRF models, this is referred to as *not-novel* view synthesis.

EpiMISR is not optimized on a per-scene basis, but rather uses a training set to learn the function needed to perform image fusion with an arbitrary geometry for the SR task in a supervised way. As shown in the high-level overview in Fig. 1, EpiMISR consists of three main modules, named SISR-FE, CAP and MIFF, which create SR features, sample them along epipolar lines and fuse them, and will be detailed in the following sections. Notice that EpiMISR also computes a super-resolved image from only the target view, called $I^{\text{SISR}}$. We found that a loss function optimizing the fidelity of both the SISR and MISR outputs with respect to the HR ground truth, such as

$$L = \mathcal{L}\left(I^{\text{MISR}}, I^{\text{HR}}\right) + \alpha\mathcal{L}\left(I^{\text{SISR}}, I^{\text{HR}}\right) \tag{2}$$

provided more stable performance over a variable range of available views and ensured that the degenerate case of a single view ($V = 0$) recovers the performance of the SISR backbone. In our experiments, we used the L1 loss as $\mathcal{L}$.

### 3.1 SISR-FE MODULE

The SISR-FE (Single Image Super-Resolution Feature Extractor) module is shared across views and its purpose is to capture strong spatial priors (local correlation and, possibly, non-local self-similarity) to extract features supported on a super-resolved image grid. Each pixel in this super-resolved grid is geometrically positioned on the camera plane associated to each particular view, but its feature vector captures the information of a neighborhood. The increased resolution with respect to the original allows finer processing by the other modules. Being part of a modular approach, SISR-FE can leverage any state-of-the-art SISR architecture by truncating the final projection to RGB space. More formally, let $I_v^{\text{LR}}$ be the $v$-th view as input of the module, its output will be a set of $C$ feature maps at $s$ times the resolution:

$$\text{SISR-FE} : I_v^{\text{LR}} \in \mathbb{R}^{H,W,3} \to F_v \in \mathbb{R}^{sH,sW,C} \qquad \forall v = 0, \dots, V \tag{3}$$

where $v = 0$ denotes the target view. We also remark that a SISR image prediction $I^{\text{SISR}}$ is obtained from $F_0$ via projection of features to RGB values, and it is used as a basis for the multi-image residual correction estimated by the other modules.

#### 3.1.1 CAP MODULE

In order to handle potentially large geometric disparities in camera poses, epipolar geometry is employed instead of the optical flow modules commonly used in the burst SR literature. A deterministic, non-learnable module called CAP (CastAndProject) is used to implement epipolar geometry with an approximate pinhole camera model. Given a pixel on the SR target view grid, there exists an associated straight line, called the epipolar line, for each of the extra views, such that the line will intersect with the object imaged by the target pixel. The CAP module is shared across the extra views, and receives as input the camera parameters of the target view $\mathcal{P}_0$, the camera parameters of the $v$-th view $\mathcal{P}_v$ and the super-resolved feature map of the $v$-th view $F_v$ to compute the epipolar features $E_v$.

$$\text{CAP}_{\mathcal{P}_0} : (F_v, \mathcal{P}_v) \to E_v \in \mathbb{R}^{P,sH,sW,C} \qquad \forall v = 1, \dots, V \tag{4}$$

The epipolar features tensor $E_v$ denotes the epipolar lines for view $v$ sampled at $P$ locations, for each pixel and feature in the target view.

The purpose of this module is to build the tensor $E_v$ so that the following MIFF module can efficiently scan the epipolar line in search of features in the extra views that match the feature in the

target view at each target pixel position, thus effectively exploiting inter-view information. For each pixel in the target view, CAP casts a ray in the 3D space passing through the center of the target camera and the selected pixel (using $\mathcal{P}_0$). Along this ray, $P$ points are sampled. For each sampled point, the module computes the projection point onto the image plane of the extra view (using $\mathcal{P}_v$). As the obtained coordinates can be non-integer, the module bicubically resamples the super-resolved feature maps $F_v$ at the correct coordinates. This also highlights the importance of having features $F_v$ on a super-resolved grid to properly account for fine details. The module also generates a boolean mask to flag invalid projected points that are outside the feature map or behind the extra camera. We also note that CAP samples points hyperbolically along the ray, so that the points are equally spaced when projected on the image planes.

### 3.1.2 MIFF MODULE

The MIFF (Multi Image Feature Fusion) module receives as input the epipolar feature tensors $E_1, \ldots, E_V$ returned by the CAP module, containing features from the extra views, warped and aligned to the target view. Its task is to aggregate them to return a residual correction to the SISR image of the target view that accounts for the information of the other views.

$$\text{MIFF} : (F_0, \{E_1, \ldots, E_V\}) \rightarrow \Delta I^{\text{SR}} \in \mathbb{R}^{sH, sW, 3} \tag{5}$$

The final super-resolved version of the target view is then obtained by:

$$I^{\text{MISR}} = I^{\text{SISR}} + \Delta I^{\text{SR}}. \tag{6}$$

Similarly to Varma et al. (2022), we drop the classical physics-based volume integral formulation, replacing it with two transformers that aggregate the information from the extra views directly in a feature space. The two transformers work in a cascade fashion, with the first transformer aggregating the views (*view transformer*) and the second transformer aggregating the points along the ray (*ray transformer*). Using the notation from Vaswani et al. (2017), each transformer is formed by an encoder and a decoder module. We refer the reader to Fig. 1 for a detailed block diagram of the following explanation.

The encoder for the view transformer considers the sequence of $V$ epipolar feature tensors $E_v$ as input and derives joint features by means of a stack of several multihead self-attention layers, feed-forward layers and LayerNorm layers (Ba et al. (2016)). This operation is crucial as it allows for the fusion of independently computed features $E_v$ from each view. By leveraging self-attention layers we enable the network to derive more intricate and integrated joint features. Also notice that this operation is equivariant to the ordering of the views and does not depend on the specific number of views $V$ available. The output of the view transformer encoder is a sequence of length $V$ of joint features. This is provided as input to the decoder together with the super-resolved features $F_0$ of the target view. The decoder uses multiple cross-attention layers to correlate the features of the target view with those extracted from the other views. Its output summarizes the content of the views in a feature field, equivalent to the radiance field in the physics-based approach of NeRF.

Next, the ray transformer replaces the physics-backed volumetric integral to integrate the feature field over the ray. Again an encoder-decoder structure is used. The encoder performs self-attention over the sequence of $P$ ray points to mix the ray features. Then the decoder uses cross-attention between the super-resolved features $F_0$ of the target view and the output of the encoder to estimate the RGB residual image correction $\Delta I^{\text{SR}}$ that is added to the SISR image.

Notice that performing the aggregation along the ray and then along the views is not optimal. However, performing both aggregations together in a single step is too computationally demanding, hence we perform first the aggregation along the views and then along the ray.

## 4 EXPERIMENTAL RESULTS

### 4.1 EXPERIMENTAL SETTING

In this work, we address the MISR task with a supervised learning paradigm. In order to properly characterize the proposed method from an experimental standpoint, we need a setting with multiple images having relatively large disparity compared to the more conventionally studied burst SR

setting. Consequently, we use the DTU dataset (Jensen et al. (2014)), which is already known in the NeRF literature, for this new SR setting. In particular, we utilize the rectified DTU dataset[1], comprising 124 different scenes, with 49 posed views per scene, each view having $1600 \times 1200$ pixels. For reasons of computational efficiency, we first bicubically downsample the original images by a factor of 4 obtaining the $400 \times 300$ HR images from which degraded LR images are derived. We split the dataset into train, validation and test. Validation set is formed by only scene 47 while the test set is formed by scenes 3, 10, 13, 18, 30, 63, 77, 99, 103. All the other 114 scenes form the train split. From each scene, multiple input sets are extracted by selecting as the target view a random image among the 49 and then choosing the nearest $V$ images as extra views, with respect to camera centers. The number of extra views during training is $V = 7$ and, unless otherwise stated, the same number is also used for testing. The angle between the target view and the other views ranges between 11 and 33 degrees, averaging around 15 degrees, which is in line with our large disparity setting.

In our experiments, the SISR-FE module is based on the SwinIR architecture (Liang et al. (2021)) in order to be comparable with recent methods in the burst SR literature. We also present some ablations with simpler designs for SISR-FE in Sec. 4.4. The number of points sampled by the CAP module along the ray during training is $P = 256$, and, unless otherwise stated, the same number is used during testing. Finally, regarding the MIFF module, we set the number of encoder and decoder layers to 4 for both transformers.

The training pipeline of EpiMISR for the following experiments consists of two steps. First, we pretrain the SISR-FE module and its RGB projection as a SISR neural network on the DIV2K dataset from Agustsson & Timofte (2017), and finetune it on the DTU dataset. Then the whole EpiMISR architecture is trained end-to-end for the MISR task, using the loss in Eq. 2 with $\alpha = 1$.

We employ the Adam optimizer for the end-to-end optimization of EpiMISR. The SISR-FE module is frozen to the pretrained weights for the first 350 iterations to train the sole MIFF module and stabilize the training, followed by an additional 150 iterations to finetune the whole network. The learning rate is linearly warmed up for the first 60 epochs starting with $10^{-6}$ up to $10^{-4}$. A multi-step scheduler halves it at epochs $150, 250$. For the final 150 epochs, the learning rate is set to $10^{-5}$ and further halved at epochs $80, 120$. We train on four A100 GPUs for about 7 days.

We compare the proposed technique to a number of state-of-the-art approaches for multi-image super-resolution in the literature. However, we remark that our setting with relatively large parallax and free camera positions is new and different from existing settings in the super-resolution literature. The closest match is the burst SR literature, which however only considers small disparities and does not use camera poses. We consider BSRT (Luo et al. (2022)) as the state-of-the-art for the burst SR literature, and DBSR (Bhat et al. (2021)) as additional baseline. The NeRF literature has recently published the NeRF-SR method by Wang et al. (2022a). We consider this method as an interesting additional point of reference which follows the NeRF methodologies and explicitly uses camera poses. However, NeRF-SR follows a different settings as it is concerned with novel view synthesis at a higher resolution rather than not-novel view enhancement and it does not follow the supervised learning paradigm. A recent preprint by Han et al. (2023) proposes Super-NeRF, but it has not been tested due to the lack of publicly available code. Besides, its setting is also different because, similarly to NeRF-SR, it does not follow the supervised learning paradigm, it focuses on novel view synthesis and, moreover, it optimizes for perception metrics and not for distortion. All methods in our comparisons have been retrained using the authors' code and following the same pretraining procedure of EpiMISR. The number of epochs for their training has been chosen to maximize their performance on a validation set. A minor modification has been made to the burst methods to use RGB images instead of RAW mosaiced images.

## 4.2 Main Experiment

Table 1 reports our main results on the DTU dataset for a $4\times$ SR factor. For quantitative evaluation, we use PSNR as quality metric and LPIPS, SSIM and BRISQUE as perceptual metrics[2]. Metrics are computed after cropping 16 pixels on each side to avoid border effects. It can be noticed that some

---

[1]third light setting, as it is the most uniform

[2]We remark that all methods, except NeRF-SR, optimize for distortion rather than perception, see Blau & Michaeli (2018) for distortion vs. perception tradeoff.

Table 1: Quantitative results for MISR on DTU dataset.

|  |  | No. Params | PSNR ↑ | BRISQUE ↓ | LPIPS ↓ | SSIM ↑ |
|---|---|---|---|---|---|---|
|  | **EpiMISR** | 23.30M | **28.60** | 41.34 | **0.11** | **0.87** |
|  | BSRT (Luo et al. (2022)) | 20.56M | 27.84 | 44.34 | 0.13 | 0.85 |
| 4× | DBSR (Bhat et al. (2021)) | 12.91M | 26.36 | 50.08 | 0.20 | 0.80 |
|  | NeRF-SR (Wang et al. (2022a)) | 1.19M | 23.17 | 34.64 | 0.32 | 0.64 |
|  | SwinIR (Liang et al. (2021)) | 14.70M | 26.87 | 45.68 | 0.17 | 0.82 |

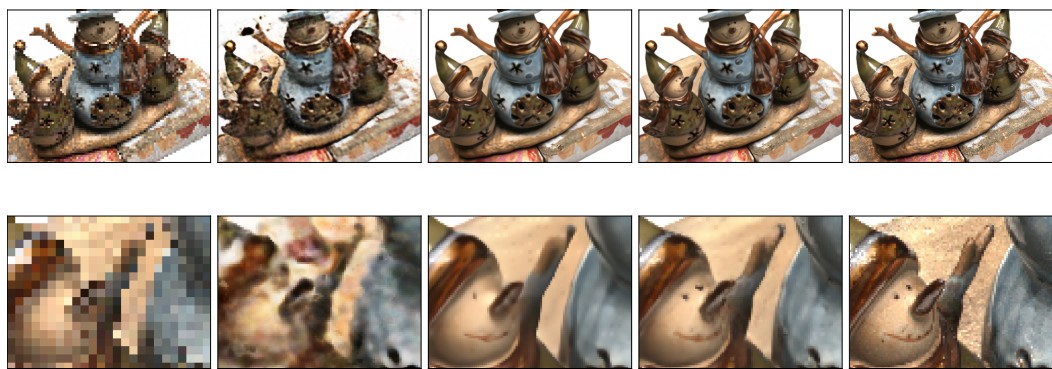

Figure 2: DTU scene 3 with 4× scale factor. From left to right: LR nearest neighbours interpolation (19.31 dB), NeRF-SR (19.75 dB), BSRT (23.60 dB), EpiMISR (24.43 dB), HR ground truth.

multi-image methods with weak spatial priors struggle to improve over the SISR result of SwinIR. As a sanity check, we tested but not reported in the table the SISR performance of EpiMISR after all the finetuning procedures, and saw that it is just marginally above the reference SwinIR results (26.96 dB), confirming that improvements actually come from the use of multiple images. The state-of-the-art from the burst SR literature (BSRT) shows a significantly lower PSNR of about 0.8 dB compared to EpiMISR, highlighting the importance of explicitly modeling the problem geometry at the core of our model rather than relying on optical flow. NeRF-SR does not show competitive performance, which is expected for several reasons: i) it targets the novel view synthesis setting; ii) it is optimized on a per-scene basis, thus not being able to learn powerful image priors from training data; iii) it is a much smaller model. The BRISQUE metric has an unusual behavior on the NeRF-SR images, as its good result is not confirmed by the other metrics nor by visual inspection. Fig. 2 shows a qualitative comparison between the proposed method and the other baselines. It can be noticed that EpiMISR provides more accurate details.

### 4.3 Number of views and Number of points along the rays

In this section, we study the impact of two important parameters of the proposed method, namely $V$, the number of extra views, and $P$ the number of points along the ray.

It can be expected that increasing the number of views $V$ allows to integrate extra information and increase the quality of the SR image. However, diminishing returns are expected, especially for extra views with very large disparity. Fig. 3a reports the PSNR of the SR image for different number of views used by the super-resolution process. Images are added by expanding the neighborhood of available views around the target, so they are progressively farther or more angled with respect to the target. We notice that only a marginal improvement is obtained increasing from 8 to 16 views. Regarding views, we also remark that EpiMISR can process an arbitrary number of input views with an arbitrary ordering, as its operations are invariant in that dimension.

The number of ray points $P$ determines the density of the feature field that takes the place of the radiance field in our model. This parameter is strictly tied to the resolution of the images and the scene characteristics, and its sampling should be fine enough to capture the fine details of the scene. Fig. 3b shows that a too small value of $P$ has a significant impact on SR quality, while performance saturates beyond the chosen value of $P = 256$.

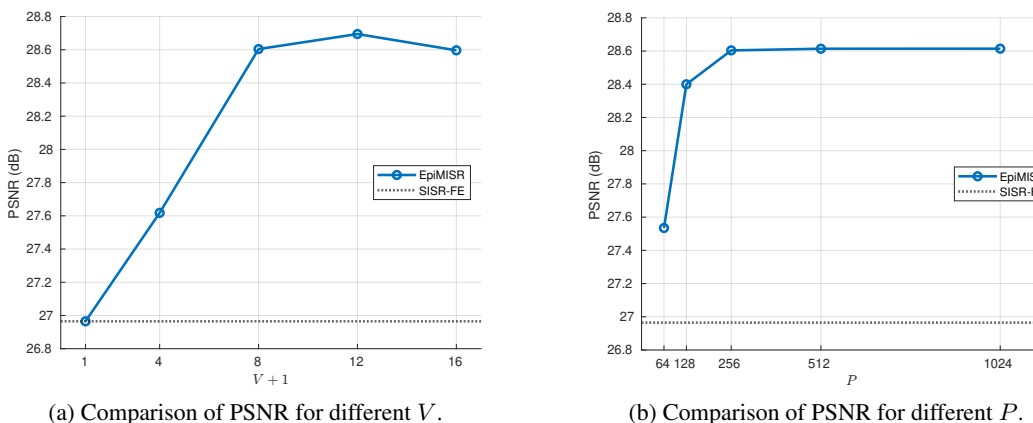

(a) Comparison of PSNR for different $V$.      (b) Comparison of PSNR for different $P$.

Figure 3: PSNR with respect to $V$ and $P$.

Table 2: Comparison of different SISR-FE modules in terms of MISR and SISR performance.

| SISR-FE Module | No. Params | PSNR (SISR) ↑ | BRISQUE ↓ | LPIPS ↓ | SSIM ↑ |
|---|---|---|---|---|---|
| **SwinIR** | 14.85M | 28.60 (26.96) | 41.34 | 0.11 | 0.87 |
| RLFN (Kong et al. (2022)) | 0.86M | 28.05 (26.38) | 42.49 | 0.12 | 0.86 |
| Bicubic + conv3×3 | 7.94k | 25.73 (24.13) | 50.79 | 0.23 | 0.79 |
| Bicubic + conv1×1 | 1.80k | 24.56 (24.04) | 52.07 | 0.27 | 0.76 |

## 4.4 SISR-FE ABLATION

The EpiMISR modular design allows to decouple the fusion of multiple images using the 3D geometry from the super-resolved feature extraction, which can leverage advances in SISR methods or be tuned for the desired complexity. In this section, we present some MISR results using different SISR-FE modules in order to study its impact on overall performance. Results are shown in Table 2. Unsurprisingly, the SwinIR architecture used in the main experiment provides the best performance but it is also a relatively large model. However, it is interesting to notice that the RLFN architecture by Kong et al. (2022) from the NTIRE 2022 challenge on Efficient Super-Resolution is able to still improve over BSRT with a fraction of the parameters. We also notice that bicubic upsampling followed by $1 \times 1$ RGB-to-features convolution is not sufficient to provide reasonable performance, highlighting the need for operations that capture a local context larger than 1 pixel. In fact, when bicubic upsampling is followed by $3 \times 3$ convolution the subsequent MIFF module is able to successfully exploit the local context as the overall performance increases by 1.17dB while the SISR performance stays almost the same. We also notice that the PSNR difference between the single-image and multi-image results is stable around 1.6 dB, proving that the MIFF module is relatively robust to the single-image processing.

## 4.5 SENSITIVITY ANALYSIS TO CAMERA PARAMETER ESTIMATION

Camera parameters in the DTU dataset are highly accurate as they have been obtained from a calibration procedure. One may wonder how performance of EpiMISR is affected by the accuracy of camera parameters. To this end, we use the state-of-the-art HLOC algorithm from Sarlin et al. (2019) to infer poses from the LR images alone. We report a MISR PSNR of $28.10$ dB, which is degraded from the result with accurate poses but still superior to BSRT which does not need that information, confirming that a large part of the improvement actually comes from the correct 3D geometry modelling.

More in detail, a sensitivity analysis to perturbations of the extrinsic camera parameters is shown in Fig. 5. It shows the PSNR achieved when the 6-D DTU pose is perturbed to simulate uncertainty. A diagonal zero-mean Gaussian with parameter $\sigma_{\text{translation}}$ is used to perturb the translational components. A simple symmetric distribution over $SO(3)$ with parameter $\sigma_{\text{rotation}}$ is used to perturb the rotational component. As Fig. 5 shows, the performance of EpiMISR degrades in higher noise poses regime, but it is still superior to BSRT in a lower noise regime and, overall, it exhibits a stable trend.

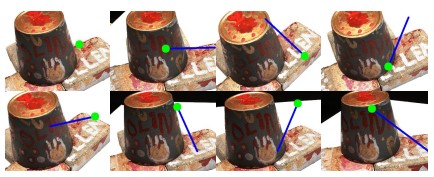 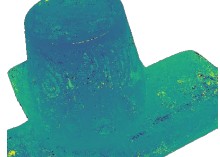

(a) Input image set with epipolar lines.

(b) Strip aligned against Ray Transformer attention.

(c) Depth map.

Figure 4: An example of depth map generation.

Finally, we remark that camera parameter estimation from LR images performed disjointly from the SR process is clearly suboptimal. Future work may significantly improve the results by designing joint methods that correct an initial pose estimation while performing super-resolution, similarly to what is done by NeRF methods for in-the-wild images (Martin-Brualla et al. (2021)).

## 4.6 ANALYSIS OF RAY ATTENTION

In this section, we present an interpretation of the attention map generated by the ray transformer within the MIFF module as a depth map.

Fig. 4a illustrates a typical input image set. The first image is the target view, while the subsequent $V = 7$ images are the extra views. Let us fix the pixel to be superresolved in the target image. The CAP modules casts a ray through this pixel and projects it onto the other views. This process yields samples along the epipolar lines, which are collected to form a "strip" of dimensions $P \times (V + 1)$, depicted in Fig. 4b (depiction is in RGB space instead of feature vectors). There are $P$ columns because the CAP module samples $P$ points along the epipolar lines, and there are $V+1$ rows because there are $V + 1$ epipolar lines. It is worth noting that the first row comprises repeated instances of the same pixel, as the epipolar line collapses to a single point in the target view. Thanks to the property of epipolar geometry, there is a region along the strip, which we will call "strip align-

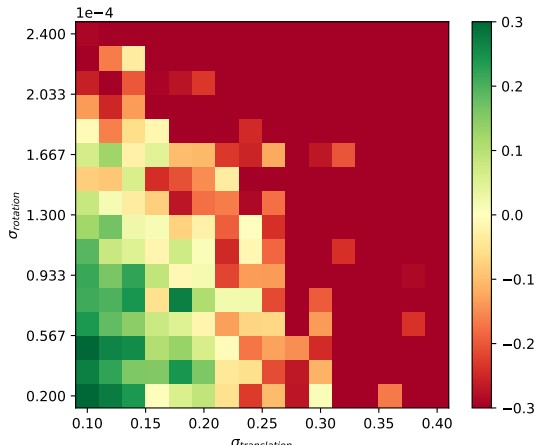

Figure 5: EpiMISR PSNR gain (dB) over BSRT for different noise regimes on camera poses for a single test image.

ment region", where all the views are imaging the same 3D point, hence the sampled feature map should report similar information. The attention weights generated by the ray transformer are also visualized in Figure 4b and we can see they reach their maximum in the alignment region, meaning that the MIFF module has identified the correspondences across all extra images. Moreover, the position of the maximum attention weight provides an estimate of the depth of the object imaged by the selected pixel in the target view. A noisy depth map for all pixels can be extracted in this unsupervised way and is visualized in Fig. 4c.

## 5 CONCLUSIONS & FUTURE WORKS

We presented a novel setting for multi-image super-resolution which addresses the case of sets of images with arbitrary camera placements, possibly with large disparities. The explicit use of epipolar geometry in the design of the super-resolution algorithm allows to achieve substantial improvements over existing methods that rely on optical flow. Future work will focus on increasing the robustness to uncertain camera parameters and moving beyond the pinhole camera to model more complex degradation effects.

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
