# OpenReview forum: "Learning Epipolar Feature Fields for Multi-Image Super-Resolution"
_ICLR.cc/2024/Conference — Submitted to ICLR 2024_

### Official Review · Reviewer_de1T · 2023-10-21

**Soundness:** 3 good
**Presentation:** 3 good
**Contribution:** 2 fair
**Rating:** 6
**Confidence:** 5

**Summary:**

The author proposed a multi-image super-resolution algorithm. The algorithm can handle input images with large parallax by combining neural processing and epipolar geometry. The proposed method achieves promising numerical and visual results.

**Strengths:**

The overall pipeline of multiple image SR with large parallax is reasonable.

The method achieved SOTA scores on objective metrics and the result is visually promising.

**Weaknesses:**

* Novelty:
I don't agree with the statement in 2.2 “this is the first work tackling the problem of generic multi-image super-resolution”. In paper [1], the framework of multi-image fusion has been proposed, and the example in section 4.1 shows the super-resolution application.

Registration and fusion neural features along the epipolar line can also be found in [1, 2, 3].

So this novelty of the proposed method is compromised. The authors should properly cite the relevant papers, make comparisons and rephrase the contributions. Will re-evaluate the novelty based on the revision.

* The description of CAP module in section 3.1.1 is ambiguous. I suggest illustrating the process with diagrams.

* The experiment section only visualizes one result in Fig. 2. To avoid the impression of handpicking, more results should be demonstrated.

* The experiments were only carried out on one dataset, so it is difficult to analyze the generalization ability of the model to other data. This is important especially in case the proposed method was claimed to be generic. The authors are suggested to evaluate the model on other data if possible.

[1] Huang, Qian, Minghao Hu, and David J. Brady. "Array Camera Image Fusion using Physics-Aware Transformers." Journal of Imaging Science and Technology 66 (2022): 1-14
[2] He, Yihui, et al. "Epipolar transformers." Proceedings of the ieee/cvf conference on computer vision and pattern recognition. 2020.
[3] Wang, Xiaofeng, et al. "MVSTER: Epipolar transformer for efficient multi-view stereo." European Conference on Computer Vision. Cham: Springer Nature Switzerland, 2022.

**Questions:**

P = 256 in experimental setting 4.1. But sometimes epipolar line can intersect with more than 256 pixels when the image resolution is 400x300. Please elaborate on how 256 points are sampled under this situation.

The parallax in NERF dataset may be extreme since the occlusions between the camera views are magnified. How does the model perform in regions with substantial occlusions?

Why do you use BRISQUE scores when you have GT and you cannot justify it?

---

> ### Author Response · Authors · 2023-11-16
> **Response to reviewer de1T**
>
> We thank the reviewer for their assessment of our submission and the suggestion they provide. In the following, we clarify the concerns on a point-by-point basis.
>
> > Novelty: I don't agree with the statement in 2.2 “this is the first work tackling the problem of generic multi-image super-resolution”. In paper [1], the framework of multi-image fusion has been proposed, and the example in section 4.1 shows the super-resolution application.Registration and fusion neural features along the epipolar line can also be found in [1, 2, 3].So this novelty of the proposed method is compromised. The authors should properly cite the relevant papers, make comparisons and rephrase the contributions. Will re-evaluate the novelty based on the revision.
>
> We thank the reviewer for the interesting references which we agree constitute relevant material that we added to the background. However, we believe there are significant differences with respect to the setting and methodology of our work. In particular, none of them addresses the multi-image super-resolution problem. Concerning section 4.1 of [1] we respectfully disagree with the statement that it deals with a MISR problem similar to ours. In fact, that experiment is more akin to pansharpening where only two images (also taken from the same vantage point) are used where a high-res monochromatic one is used to guide a resolution enhancement of the second one. Other differences with respect to ref. [1] concern their use of a voxelization approach which introduces undesirable approximations. Other works, such as [2,3], tend to process the multiple images in pairs rather than in a fully-joint way as in our proposed method, thus limiting the capability to derive joint features, ensure consistency or robustness. In particular, [2] works in pairs and then uses a criterion to select the best pair (sec. 4.1 of [2]: “During testing, we can choose different neighboring views as the source view and select the prediction with the highest confidence (i.e., highest peak on the heatmap”)); [3] also works in pairs and then performs a non-parametric aggregation (sum). In light of all these differences, we believe that our work has an interesting novel approach that is not currently found in the literature.
>
> > The description of CAP module in section 3.1.1 is ambiguous. I suggest illustrating the process with diagrams.
>
> Due to space constraints, it is difficult to include further diagrams. However, a sketch of its behavior is shown in Fig.1 and also Fig.4a. We also tried to improve the textual description to avoid ambiguities.
>
> > The experiment section only visualizes one result in Fig. 2. To avoid the impression of handpicking, more results should be demonstrated.
>
> The appendix now includes an extended set of visualizations, and an example of a failure case.
>
> > The experiments were only carried out on one dataset, so it is difficult to analyze the generalization ability of the model to other data. This is important especially in case the proposed method was claimed to be generic. The authors are suggested to evaluate the model on other data if possible.
>
> We now present an expanded set of results in the supplementary material on the 1023 test scenes from the dataset used by the IBRNet paper (Google Scanned Objects), and the entire LLFF dataset, which confirm the observed improvement over BSRT.
>
> > P = 256 in experimental setting 4.1. But sometimes epipolar line can intersect with more than 256 pixels when the image resolution is 400x300. Please elaborate on how 256 points are sampled under this situation.
>
> The value of P, number of points along a ray, is a scene-dependent hyperparameter which also depends on the resolution of the image. More critically, it also depends on the spatial width of the features to be merged from the different views. In Sec. 4.6 (Fig. 4b) we show that the attention weights are significant only for the samples where there is a match in the features. Hence, for a given ray, we are more concerned to have a high value of P to sample densely enough, rather than cover all the pixels. Also, we remind that the features at the P sampled points are obtained via bicubic interpolation from the feature field on the pixel grid to capture their non-integer location as best as possible. The experiment in Fig. 3b shows that there is no gain in increasing the value of P beyond 256 for the DTU dataset and our resolution.
>
> Continues in next comment.

---

> ### Author Response · Authors · 2023-11-16
> **Response to reviewer de1T (pt.2)**
>
> > The parallax in NERF dataset may be extreme since the occlusions between the camera views are magnified. How does the model perform in regions with substantial occlusions?
>
> As also addressed in the response to a comment from reviewer usbg, we believe that further works could be dedicated to addressing occlusions more explicitly. However, we believe that the use of the view transformer makes the method robust to occlusions since views with inconsistent features due to occlusions are rejected by the attention mechanism. We back up this claim by showing an experiment on the suggested fern scene from the LLFF dataset where EpiMISR still outperforms the state-of-the-art BSRT.
>
> >  Why do you use BRISQUE scores when you have GT and you cannot justify it?
>
> As it is a standard metric used in many works, we report it to aid future comparisons even if we do have the ground truth image.

---

> > ### Comment · Reviewer_de1T · 2023-12-01
> >
> > After reading the authors' responses, I am not opposed to the statement that it is a new method to deal with multi-image superresolution problems. As the reviewer usbg and I pointed out, however, using the epipolar geometry in multi-image neural processing is not novel. I understand the implementation can be new and specific to the application, but the novelty of the proposed method is inevitably compromised.
> >
> > The responses to my other concerns are satisfactory.
> >
> > Overall I would like to change my rating to borderline accept, considering the technical novelty and soundness of the experiments. With that being said, the paper if accepted should properly cite the related work to acknowledge the previous effort on multi-image processing and usage of epipolar geometry.

---

### Official Review · Reviewer_usbg · 2023-10-30

**Soundness:** 2 fair
**Presentation:** 2 fair
**Contribution:** 2 fair
**Rating:** 3
**Confidence:** 5

**Summary:**

The paper introduces a method, known as EpiMISR, for multi-image super-resolution by integrating complementary elements from multiple neighboring images. EpiMISR leverages the epipolar geometry inherent in the image acquisition process and employs transformer-based processing of radiance feature fields. This approach improves the performance of MISR methods, particularly when dealing with large disparities in low-resolution images. Once the super-resolution model is trained, it can be applied to a new scene with an arbitrary number of views. However, the method does not have the capability to render novel views.

**Strengths:**

+ The paper proposes a novel method for multi-image super-resolution.
+ The paper is articulated in a clear and concise manner, making the method straightforward to comprehend.

**Weaknesses:**

1. Novelty: While the use of epipolar geometry is not new and has been widely used in light field super-resolution [1] and generalizable NeRF [2], the main pipeline of this paper appears to be a combination of single-image super-resolution methods and light field neural rendering [3]. The paper’s primary contribution is unclear.
[1] Zhang, Shuo, Song Chang, and Youfang Lin. "End-to-end light field spatial super-resolution network using multiple epipolar geometry." IEEE Transactions on Image Processing 30 (2021): 5956-5968.
[2] Suhail, Mohammed, et al. "Generalizable patch-based neural rendering." European Conference on Computer Vision. Cham: Springer Nature Switzerland, 2022.
[3] Suhail, Mohammed, et al. "Light field neural rendering." Proceedings of the IEEE/CVF Conference on Computer Vision and Pattern Recognition. 2022.

2. View Consistency: View consistency is indeed crucial for multi-image super-resolution, differing from single-image super-resolution. If the method can ensure consistency among all super-resolution images, it would be beneficial for downstream tasks such as novel view synthesis and 3D reconstruction. However, the paper does not evaluate the consistency of the produced super-resolution images.

3. More Datasets: The paper claims that the method can be used for an arbitrary scene with an arbitrary number of views with an arbitrary geometry. However, only 9 scenes in the DTU dataset are tested, which may not be sufficient to demonstrate the method’s effectiveness, especially in super-resolution tasks. IBRNet[1] has collected multiple multi-view datasets, which contain more than 1000 scenes. Testing on this dataset would provide more convincing results.
[1] Wang, Qianqian, et al. "Ibrnet: Learning multi-view image-based rendering." Proceedings of the IEEE/CVF Conference on Computer Vision and Pattern Recognition. 2021.

4. Occlusion: The method’s direct fusion of multi-view features in the MIFF module may introduce artifacts due to occlusions. It’s unclear how the method deals with occlusion problems. While the occlusions in DTU scenes are not complex, allowing the method to outperform some single-image super-resolution methods, it’s uncertain whether it can be applied to complex scenes and achieve similar performance, such as the fern scene in the LLFF dataset.
5. Missing a reference.
Huang, Xin, et al. "Local implicit ray function for generalizable radiance field representation." Proceedings of the IEEE/CVF Conference on Computer Vision and Pattern Recognition. 2023.

**Questions:**

I hope the author can address my concerns mentioned in the weaknesses. Additionally, I have a few more questions and suggestions:

1. Is it possible to apply epipolar geometry to dynamic scenes? Additionally, epipolar geometry is dependent on pose calibration. Could you clarify why methods based on epipolar geometry are considered superior to flow-based methods?

2. Is the method dependent on bicubically downsampling? If given low-resolution images captured by a camera as input, can the method produce super-resolution images?

3. The caption of the pipeline could be more detailed for better understanding.

---

> ### Author Response · Authors · 2023-11-16
> **Response to reviewer usbg**
>
> We thank the reviewer for their assessment of our submission and the suggestion they provide. In the following, we clarify the concerns on a point-by-point basis.
>
> > Novelty: While the use of epipolar geometry is not new and has been widely used in light field super-resolution [1] and generalizable NeRF [2], the main pipeline of this paper appears to be a combination of single-image super-resolution methods and light field neural rendering [3]. The paper’s primary contribution is unclear. [1] Zhang, Shuo, Song Chang, and Youfang Lin. "End-to-end light field spatial super-resolution network using multiple epipolar geometry." IEEE Transactions on Image Processing 30 (2021): 5956-5968. [2] Suhail, Mohammed, et al. "Generalizable patch-based neural rendering." European Conference on Computer Vision. Cham: Springer Nature Switzerland, 2022. [3] Suhail, Mohammed, et al. "Light field neural rendering." Proceedings of the IEEE/CVF Conference on Computer Vision and Pattern Recognition. 2022.
>
> We thank the reviewer for suggesting reference [1] that we agree is relevant to the problem and that we added to the background material along with the already cited [2,3]. However, we believe there are significant differences that set our setting and methodology apart from those works. Concerning reference [1], light-field super-resolution can be considered a special case of our setting with a very restrictive grid-like, fixed and well-known geometry and small disparities. This allows the use of simpler techniques for image fusion and does not have requirements of robustness as stringent as the large disparity setting. As an example, our setting is required to deal with potential occlusions (also see the answer below about this specific point), missing correspondences due to a large angle with respect to the target camera, as well as non-Lambertian surfaces (while light field SR can safely use Lambertian approximations due to the small disparities). To further clarify this point, the methodology presented in [1] exploits the grid-like acquisition setup of light-field cameras to construct stacks of views along 4 fixed directions, allowing convenient registration and processing via 3D convolutions. This approach does not clearly generalize to the large disparity setting we study in this paper. Concerning robustness, our setting calls for the use of transformers along epipolar lines (which is not done in [1]) in order to properly match features between the target view and all the other views in an input-dependent fashion that can thus suppress mismatches, occlusions and analyze scene variations.
> Concerning references [2] and [3] they do not address the super-resolution task, and are also concerned with novel view synthesis which is outside our scope. The idea of a generalizable NeRF (transductive learning) rather than per-scene optimization (inductive learning) is a natural approach when dealing with inverse problems and can be found in other works, including some referenced in our paper. This is because, contrary to the view synthesis task, a severely ill-posed inverse problem like super-resolution requires learning strong data priors from training data, and per-scene optimization is well-known to yield suboptimal results (also see our comparisons with NeRF-SR ). Therefore, we do not really regard a transductive approach to our problem as a novel contribution of ours but rather a fundamental prerequisite. Our contribution lies in proposing a state-of-the-art architecture that is capable of robustly addressing a setting that is more challenging than the restrictive existing ones (burst, lightfield, …). We thus believe that the novelty of the method should be measured against the state-of-the-art of what can be currently applied to the large-disparity MISR problem, for which the closest approach is the BSRT from the burst SR literature.
>
> Continues in next comment.

---

> ### Author Response · Authors · 2023-11-16
> **Response to reviewer usbg (pt.2)**
>
> > View Consistency: View consistency is indeed crucial for multi-image super-resolution, differing from single-image super-resolution. If the method can ensure consistency among all super-resolution images, it would be beneficial for downstream tasks such as novel view synthesis and 3D reconstruction. However, the paper does not evaluate the consistency of the produced super-resolution images.
>
> The reviewer raises an interesting point about view consistency which we think deserves clarification. The setting we study is that of not-novel view synthesis, more akin to the standard super-resolution literature rather than the radiance field reconstruction literature. As such, it can be misleading to think of consistency as a global requirement, which would be desirable in super-resolving the entire radiance field for novel view synthesis applications. Instead, in our case we are only concerned with generating details that are consistent with the LR observations of the target view we want to super-resolve. The transformers used as building blocks of our method implicitly ensure that only consistent information is borrowed from the other views via the attention mechanism. In order to experimentally measure consitency, we include an additional result in the supplementary material, about the L2 norm of the error between the LR target image and the SR target image when degraded to LR. This assesses how different methods to solve the inverse problem ensure consistency with the observations. Moreover, we repeat it to super-resolve all the images in the scenes to simulate the case in which one wants to super-resolve to entire image set (possibly for further downstream tasks) rather than just one view.
>
> > More Datasets: The paper claims that the method can be used for an arbitrary scene with an arbitrary number of views with an arbitrary geometry. However, only 9 scenes in the DTU dataset are tested, which may not be sufficient to demonstrate the method’s effectiveness, especially in super-resolution tasks. IBRNet[1] has collected multiple multi-view datasets, which contain more than 1000 scenes. Testing on this dataset would provide more convincing results. [1] Wang, Qianqian, et al. "Ibrnet: Learning multi-view image-based rendering." Proceedings of the IEEE/CVF Conference on Computer Vision and Pattern Recognition. 2021.
>
> We now present an expanded set of results in the supplementary material by including all the 1023 test scenes from the IBRNet dataset (Google Scanned Objects) and the entire LLFF dataset. These results confirm the improvements over the state-of-the-art BSRT method, sometimes even reporting a more substantial gain compared to the experiment on the DTU data.
>
> > Occlusion: The method’s direct fusion of multi-view features in the MIFF module may introduce artifacts due to occlusions. It’s unclear how the method deals with occlusion problems. While the occlusions in DTU scenes are not complex, allowing the method to outperform some single-image super-resolution methods, it’s uncertain whether it can be applied to complex scenes and achieve similar performance, such as the fern scene in the LLFF dataset.
>
> We think that occlusions are a tricky topic that deserves further future investigation. Nevertheless, we think that the proposed method is robust to occlusions thanks to the use of view transformers when fusing views. Since the transformer implements an input-dependent function, it can reject inconsistent contributions from occluded views by computing small attention weights. We back up this claim by showing an experiment on the suggested fern scene where EpiMISR still outperforms the state-of-the-art BSRT.
>
> > Missing a reference. Huang, Xin, et al. "Local implicit ray function for generalizable radiance field representation." Proceedings of the IEEE/CVF Conference on Computer Vision and Pattern Recognition. 2023.
>
> Thanks for the suggestion, we included the relevant work in the background material.
>
> Continues in next comment.

---

> ### Author Response · Authors · 2023-11-16
> **Response to reviewer usbg (pt.3)**
>
> > Is it possible to apply epipolar geometry to dynamic scenes? Additionally, epipolar geometry is dependent on pose calibration. Could you clarify why methods based on epipolar geometry are considered superior to flow-based methods?
>
> Extending the work to dynamic scenes is an interesting direction that we are currently considering. However, it has its own challenges and first requires knowing how the baseline method works on static scenes, which is the goal of this paper. For a static scene, the epipolar geometry is the more accurate geometric model of the scene. As we discuss in the paper, optical flow is locally translational on the camera plane and thus is accurate only with small parallax between cameras, such as in burst photos. An extension to dynamic scenes would require supplementing the epipolar geometry with a model of temporal change. Indeed, a hybrid model using both epipolar geometry for the static parts and optical flow for change could be an interesting idea to study, but this is a future step that needs to rely on the foundations set in this paper.
>
> > Is the method dependent on bicubically downsampling? If given low-resolution images captured by a camera as input, can the method produce super-resolution images?
>
> It is a common approach in the super-resolution literature to consider bicubic downsampling as a standard benchmark when the focus of the contribution is not the degradation model, but rather architectural improvements. Indeed, there is nothing constraining the proposed method from being trained with different degradation models. For LR camera images that have undergone unknown degradations, the best results would be obtained by pairing the proposed method with some kernel estimation technique or ideas from the blind SR literature. This is interesting but it is beyond the scope of this paper which focuses on benchmarking the architecture design.
>
> > The caption of the pipeline could be more detailed for better understanding.
>
> We improved the caption to provide a high-level description of the pipeline.

---

### Official Review · Reviewer_Cf89 · 2023-11-01

**Soundness:** 3 good
**Presentation:** 3 good
**Contribution:** 3 good
**Rating:** 8
**Confidence:** 5

**Summary:**

The paper proposed a novel MISR technique by employing epipolar constraints in classical multi-view geometry. There were similar MISR tasks where the multiple images were obtained from adjacent frames in video, burst images, and stereo images, which means that the multiple images have small disparity. The authors claim that the proposed methods can super-resolve images with potentially wider disparity. They provided few output results and quantitative results.

**Strengths:**

- Overall, it is a well-written and easy to read paper.
- The authors clearly differentiated the proposed work compared with existing MISR works about the definition and technical novelty.
- They also provided experimental results of the existing SOTA methods.
- They also provided degree of disparity in the experimental images (a.k.a. parallax angle)
- Technical novelty is on applying epipolar constraint to the learning models, since to the deep network, information which doesn't satisfy the epipolar constraint can be regarded as noise to the problem setting.

**Weaknesses:**

The paper already provides contents that I was about to challenge. But I suggest that, as it is image related works, the authors could provide more qualitative results (sample images).

**Questions:**

No questions

---

> ### Author Response · Authors · 2023-11-16
> **Response to reviewer Cf89**
>
> We thank the reviewer for their positive assessment. Please have a look at the supplementary material that we uploaded to address the reviewers’ suggestions. Among other things, we also provide expanded qualitative results, as suggested.

---

### Official Review · Reviewer_eUdX · 2023-11-06

**Soundness:** 4 excellent
**Presentation:** 4 excellent
**Contribution:** 4 excellent
**Rating:** 10
**Confidence:** 4

**Summary:**

A method for multi-image super-resolution (SR) is proposed where the input multi-images which provide the extra subpixel information are captured from a wider baseline rather than a video or very small baseline. These kind of images are typical in structure from motion settings where epipolar geometry is used to accumulate the extra information from all the images. The solution framework is divided into three main blocks. The first block does standard single image pre-existing super-resolution (stopped at feature level output and not full RGB output) followed by capture for epiploar information with reference image being one single image which needs to be super-resolved and epiploar images being all other images. The epiploar information is encoded in 4D tensor. Lastly the epiploar tensor matrix is passed through a transformer (encoder-decoder) architecture to learn a delta image which when linearly added to already existing single image super resolved image will improve it even further. Many relevant metrics are shown in results to show that the proposed multi-image super resolution outperforms SOTA in single image SR and burst SR.

**Strengths:**

The paper is very well written (infact I didn't find any grammatical mistakes which is rare in my experience), easy to understand and all concepts are clearly explained with sufficient literature review. The results analysis is also good. The paper addresses an almost unexplored territory of wide base line multi-image super resolution. The analysis is Section 4.6 is very good to relate newer concepts like attention with more traditional vision analysis.

**Weaknesses:**

1. One of the missing parts in the results section is what happens when the selection of multi-view images involves widening the baseline between images, while keeping the number same. In other words if the selected images are more spread out, then how does the method perform. This result would have been like a test of breaking point of the method with respect to distribution of the camera locations. Like in Figure4, the images are wide baseline but not too wide.

2. Another thing to put would be more visual results. Figure 2 just shows one result. Typically 4-5 results with challenging scenes should have been shown as SR is a very visual problem.

3. The paper doesn't talk about failure cases. Its hard to believe that there were no cases where the proposed method outperformed all SOTA in all test images.

4. I also couldn't find any numbers on run-time for training and inference.

**Questions:**

Kindly look at the weakness section and address it.

---

> ### Author Response · Authors · 2023-11-16
> **Response to reviewer eUdX**
>
> We thank the reviewer for their positive assessment. We have strived to further improve the quality of the work by uploading an appendix addressing various points raised by the reviewers.
>
> > One of the missing parts in the results section is what happens when the selection of multi-view images involves widening the baseline between images, while keeping the number same. In other words if the selected images are more spread out, then how does the method perform. This result would have been like a test of breaking point of the method with respect to distribution of the camera locations. Like in Figure4, the images are wide baseline but not too wide.
>
> An experiment with wider baseline is presented in the appendix to show a setting with challenging geometry where the performance of BSRT degrades to that of single-image SR, while the proposed method provides improvements. This more challenging geometry is created by taking the V-1 extra views that are at median distance (out of all the views available in the dataset) with respect to the distance to the target view camera center.
>
> > Another thing to put would be more visual results. Figure 2 just shows one result. Typically 4-5 results with challenging scenes should have been shown as SR is a very visual problem.
>
> The appendix now includes an extended set of visualizations.
>
> > The paper doesn't talk about failure cases. Its hard to believe that there were no cases where the proposed method outperformed all SOTA in all test images.
>
> We now present a single failure case in the appendix. This is an example where BSRT outperforms the proposed method. We also report the histogram of the difference between the PSNR of the proposed method and BSRT, showing that those failure cases are rare.
>
> > I also couldn't find any numbers on run-time for training and inference.
>
> We added details about training time to the main text of the experimental section.

---

> > ### Comment · Reviewer_eUdX · 2023-11-22
> >
> > Thanks for the rebuttal and I am satisfied with it. Also, my suggestion/intent was that you should put more results and failure cases in the main paper and not appendix.

---

### Meta-Review · Area_Chair_Y3wt · 2023-12-07

**Metareview:**

The paper introduces a method that leverages epipolar geometry to enhance image super-resolution using multi-view images. It received divergent ratings, including one strong accept, which notably seemed biased, an accept, a borderline accept, and a reject, illustrating a substantial disparity in the reviewers' opinions.

Reviewers usbg and de1T particularly emphasized that the novelty of this approach is questionable. The integration of Transformer models with epipolar geometry, though intriguing, is not a new concept and has been previously explored in other contexts. This diminishes the novelty and impact of the paper's contribution.

Another issue of the paper is its reliance on the assumption of well-calibrated cameras. This assumption could provide an unfair advantage when comparing against baseline multi-view SR methods. It also significantly simplifies real-world scenarios, potentially reducing the practical applicability and relevance of the proposed method.

Moreover, the fusion approach of Transformer models with epipolar geometry in this study is somewhat straightforward and lacks innovation. This further casts doubt on the paper's contribution to the field, especially considering the existing body of work in similar areas.

**Justification For Why Not Higher Score:**

The paper presents an approach to combine Transformer models with epipolar geometry in the context of multi-view super-resolution (SR). The reviews of the paper show significant divergence, suggesting that the paper requires major improvements. In particular, as highlighted by Reviewers usbg and de1T, the novelty of this approach is somewhat limited, as similar integrations have been explored in different contexts.

Another significant concern is the assumption of well-calibrated cameras, which could provide an unfair advantage when comparing against baseline multi-view SR methods. This assumption simplifies the real-world application scenario and could limit the application of the proposed method.

Additionally, while Transformer models and epipolar geometry are appealing concepts, their combination in this paper seems rather straightforward.

**Justification For Why Not Lower Score:**

N/A

---

### Decision · Program_Chairs · 2024-01-16

Reject